# Extension of a multiphase tumour growth model to study nanoparticle delivery to solid tumours

Barbara Wirthl[1‡], Johannes Kremheller[1‡*], Bernhard A. Schrefler[2,3], Wolfgang A. Wall[1]

**1** Institute for Computational Mechanics, Technical University of Munich, Garching b. München, Germany,
**2** Institute for Advanced Study, Technical University of Munich, Garching b. München, Germany,
**3** Department of Civil, Environmental and Architectural Engineering, University of Padova, Padova, Italy

‡ Co-first authorship.
* kremheller@lnm.mw.tum.de

**Data Availability Statement:** All relevant data are within the paper and its Supporting Information files.

**Funding:** JK, BAS and WAW gratefully acknowledge the support of the Technical

## Abstract

One of the main challenges in increasing the efficacy of conventional chemotherapeutics is the fact that they do not reach cancerous cells at a sufficiently high dosage. In order to remedy this deficiency, nanoparticle-based drugs have evolved as a promising novel approach to more specific tumour targeting. Nevertheless, several biophysical phenomena prevent the sufficient penetration of nanoparticles in order to target the entire tumour. We therefore extend our vascular multiphase tumour growth model, enabling it to investigate the influence of different biophysical factors on the distribution of nanoparticles in the tumour microenvironment. The novel model permits the examination of the interplay between the size of vessel-wall pores, the permeability of the blood-vessel endothelium and the lymphatic drainage on the delivery of particles of different sizes. Solid tumours develop a non-perfused core and increased interstitial pressure. Our model confirms that those two typical features of solid tumours limit nanoparticle delivery. Only in case of small nanoparticles is the transport dominated by diffusion, and particles can reach the entire tumour. The size of the vessel-wall pores and the permeability of the blood-vessel endothelium have a major impact on the amount of delivered nanoparticles. This extended *in-silico* tumour growth model permits the examination of the characteristics and of the limitations of nanoparticle delivery to solid tumours, which currently complicate the translation of nanoparticle therapy to a clinical stage.

## 1 Introduction

Cancer is not only a major cause of death worldwide [1] but also a complex system, in terms of its causes and effects on the body. Cancer cells divide at an uncontrollably high rate, invade healthy tissue and travel to distant sites in other organs where they form new metastases [2]. This considerably complicates the targeting of malignant cells and their complete extinguishment. Chemotherapy, along with radiation therapy and surgery, is currently one of the main

University of Munich – Institute for Advanced Study, funded by the German Excellence Initiative and the TÜV SÜD Foundation. BAS gratefully acknowledges the support of the CITO Award, Houston Methodist Research Institute, Houston, NCI U54 CA210181. The funders had no role in study design, data collection and analysis, decision to publish, or preparation of the manuscript.

**Competing interests:** The authors have declared that no competing interests exist.

common anti-tumour therapeutic approaches. It has reduced the death rates of many cancer types, thereby saving countless lives [3]. Notwithstanding this, chemotherapy in particular causes tremendous side effects, including physical but also psychological damage, due to a lack of specificity of the anti-cancer agents [4]. The problem is that it is not only diseased tissue which is exposed to the drugs: healthy organs are also attacked. This is where Paul Ehrlich developed his idea of a *magic bullet*, which selectively kills cancer cells while leaving surrounding healthy tissue undamaged [5].

Cancer research has come closer to this goal since nanoparticles were discovered. Nanoparticles are nanosized organic or inorganic materials which can be designed with different physicochemical properties, e.g. size or shape, and programmed with various biological and medical functions [6]. This allows either for an active substance (e.g. a chemotherapeutic agent) to be encapsulated in the nanoparticles, or for it to be attached to their surface [7]. Therefore, ongoing cancer research focuses on drug delivery with nanoparticles, based on the idea of the specific targeting of malignant cells by conjugation of nanoparticles with ligands of cancer-specific tumour biomarkers [8]. This leads to a more effective targeted drug delivery to cancer cells and thereby cancer attenuation, while at the same time it reduces unwanted adverse effects. But, despite numerous promising ideas and experimental results, nanoparticles do not efficaciously reach tumours in clinical practice [9].

To overcome that deficiency, a new area called *transport oncophysics* has emerged [10]. The driving idea behind it is to describe and address cancer with a focus on the physical phenomena of mass transport [10, 11]. Tumours are classified according to their transport phenotype, which is based on specific transport properties instead of merely genetic characteristics [11]. Adequately addressing transport barriers on the way from administration to the tumour is essential to overcome current limitations of nanoparticle-based drug delivery [12].

To better understand the characteristics and in particular the barriers to nanoparticle transport, and the factors which enhance or limit their clinical application, we extend our recently developed multiphase tumour growth model to also include nanoparticle transport. The model aims to predict the accumulation of nanoparticles in the tumour and allows a deeper insight into nanoparticle transport mechanisms. To this end, the previously developed multiphase tumour model [13, 14] is extended by nanoparticle transport. The multiphase tumour growth model in its original form, including the macroscopic balance equations of phases and species, was presented by Sciumè et al. [15, 16] and enhanced to include a deformable extracellular matrix (ECM) (also by Sciumè et al. [17]) as well as ECM deposition and angiogenesis by Santagiuliana et al. [18]. Mascheroni et al. experimentally validated the model equations and conducted first studies on *in vitro* drug delivery and its interaction with mechanical stress [19, 20]. Kremheller et al. [13] introduced the neovasculature as a proper additional phase and added a hybrid embedded/homogenised treatment of the vasculature [14]. These extensions now allow us to simulate and analyse the transport of nanoparticles as occurring *in vivo*.

In this study, we extend our multiphase model to include the transport of nanoparticles. As we do not restrict the model to a specific therapeutic approach at this stage, the term *nanoparticle* is used in a generic way in the remainder of this paper. The transport barriers encountered by nanoparticles on their way from administration to the tumour are shared by all nanoparticle drugs [12]. Nanoparticles used in cancer treatment could for instance be drug-loaded nanoparticles, as presented by Curtis et al. [21], or magnetic nanoparticles mediating hyperthermia, as presented by Feng et al. [22] and Nabil et al. [23]. Nevertheless, our framework could also easily be extended to include multistage delivery systems as presented by Tasciotti et al. [24] or Venuta et al. [25] and computationally modelled by Nabil et al. [26].

The remainder of the paper is structured as follows. Section 2 gives a short overview over the multiphase tumour growth model with a special focus on the extension to include nanoparticle transport. The enhanced model is then used to study different effects, and the results are presented and discussed in Section 3, before Section 4 draws a conclusion of the presented work.

## 2 Methods

### 2.1 The vascular multiphase tumour growth model

Our multiphase tumour growth model is based on Sciumè et al. [17] with the extension to a five-phase model including the vasculature as presented in Kremheller et al. [13, 14]. The model is based on the Thermodynamically Constrained Averaging Theory (TCAT) [27], which has several benefits. By using those averaging theorems to proceed from known microscale relations to the continuum scale, the model is mathematically and physically consistent. The thermodynamic analysis is also consistent between scales by satisfying the entropy inequality. A detailed introduction to TCAT can be found in Gray and Miller [27].

The model comprises the ECM as the solid phase with three fluid phases flowing in its pores: tumour cells, host cells and the interstitial fluid (IF). The vasculature is modelled as an independent porous network where blood flow takes place. Fig 1 schematically shows the different components of the multiphase model at the microscale. However, our multiphase tumour growth model is formulated at the macroscale where the phases are modelled in an averaged sense, based on volume fractions $\varepsilon^{\alpha}$ of the arbitrary phase $\alpha$. In the following equations, $t$ denotes the tumour cells, $h$ the host cells, $l$ the IF and $s$ the ECM as the solid phase. In addition, the vasculature is denoted by $v$. The sum of the volume fractions of all five phases

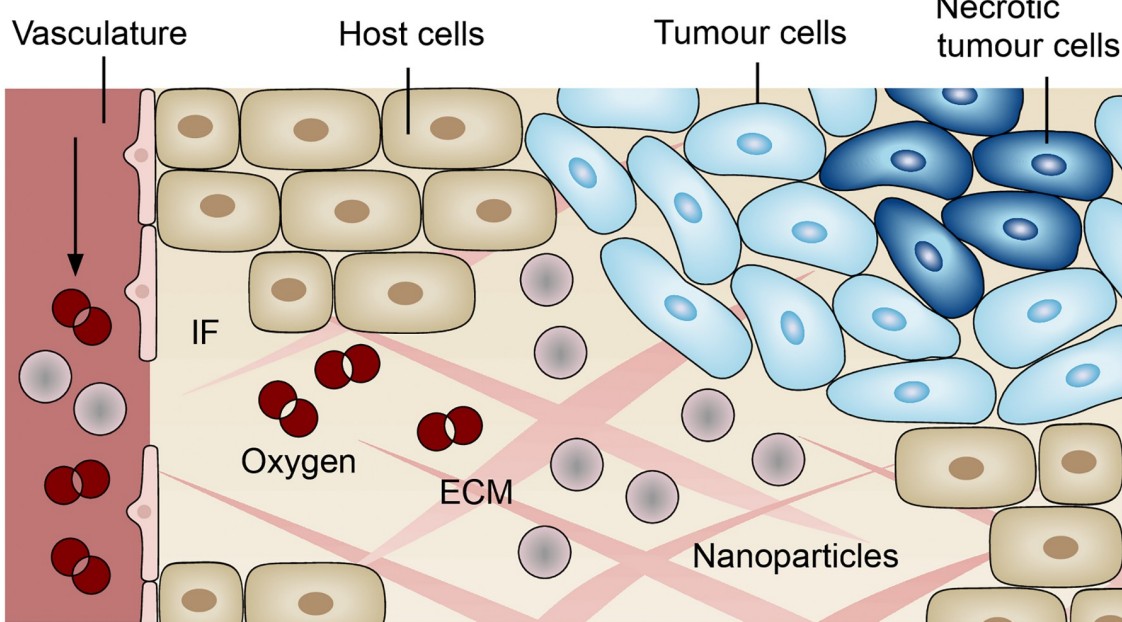

**Fig 1. Components of the multiphase tumour growth model.** The model comprises a solid phase, the ECM, three fluid phases, host cells, tumour cells and interstitial fluid (IF), and the vasculature which is modelled as an independent porous network. In addition, the phases transport species, namely necrotic tumour cells, oxygen and nanoparticles.

included in the porous media model must satisfy the equation

$$\varepsilon^s + \varepsilon^h + \varepsilon^t + \varepsilon^l + \varepsilon^v = 1. \tag{1}$$

The saturations $S^\alpha$ of tumour cells, host cells and the IF are defined as

$$S^\alpha = \frac{\varepsilon^\alpha}{\varepsilon}, \quad \text{where} \quad \alpha = h, t, l. \tag{2}$$

Furthermore, all phases can transport species. Oxygen, as a nutrient, is mainly transported by the vasculature and the IF and its mass fraction is denoted by $\omega^{n\bar{v}}$ and $\omega^{n\bar{l}}$, respectively. Sufficiently high oxygen supply leads to tumour growth. In contrast, tumour cells become necrotic when exposed to low nutrient concentrations or excessive mechanical pressure. To take this into account, tumour cells are divided into living and necrotic tumour cells, the latter being modelled as a species. The mass fraction of necrotic tumour cells in phase $t$ is denoted by $\omega^{N\bar{t}}$. In this contribution, we employ our previously developed vascular tumour growth model as a precursor to generate physically plausible results which serve as an initial condition for studying nanoparticle transport in the tumour micro-environment. The full model including all equations is described in S1 Appendix.

## 2.2 Nanoparticle transport

We now introduce nanoparticles as additional species in our tumour growth model. The mass balance equation of nanoparticles with mass fraction $\omega^{NP\bar{\alpha}}$ in phase $\alpha$ is given by

$$\varepsilon S^\alpha \frac{\partial \omega^{NP\bar{\alpha}}}{\partial t}\bigg|_X - \frac{\boldsymbol{k}^\alpha}{\mu^\alpha} \boldsymbol{\nabla} p^\alpha \cdot \boldsymbol{\nabla} \omega^{NP\bar{\alpha}} - \boldsymbol{\nabla} \cdot \left( \varepsilon S^\alpha D_{\text{eff}}^{NP\alpha} \boldsymbol{\nabla} \omega^{NP\bar{\alpha}} \right)$$

$$= \frac{1}{\rho^\alpha} \left( \sum_{\kappa \in \mathscr{J}_{c\alpha}} \overset{NP\kappa \to NP\alpha}{M} + \varepsilon^\alpha r^{NP\alpha} - \omega^{NP\bar{\alpha}} \sum_{\kappa \in \mathscr{J}_{c\alpha}} \overset{\kappa \to \alpha}{M} \right) \tag{3}$$

similar to the other species (see S1 Appendix). Herein, $\boldsymbol{k}^\alpha$ denotes the isotropic permeability of the ECM with respect to a fluid phase $\alpha$, $\mu^\alpha$ the viscosity, $p^\alpha$ the pressure and $\rho^\alpha$ the density. The effective diffusivity of nanoparticles is given by $D_{\text{eff}}^{NP\alpha}$.

We assume that nanoparticles are intravenously injected and subsequently transported by the vasculature. As mentioned above, we do not explicitly model transport in the vasculature, but rather assume a constant mass fraction of nanoparticles in the vasculature $\omega^{NP\bar{v}}$. Capillary walls are semipermeable [28] and, because of their small size, nanoparticles extravasate into the IF and therein diffuse towards the tumour. In our multiphase model, we include nanoparticles as part of the vasculature and the IF, hence $\alpha, \kappa \in \{v, l\}$. Since nanoparticles are intravenously injected and not produced by the tumour environment, the source term on the right-hand side of the equation is not present ($\varepsilon^\alpha r^{NP\alpha} = 0$).

The other mass transfer terms on the right-hand side of Eq (3) are discussed in the following. The extravasation of nanoparticles from the capillaries into the IF occurs through two different pathways: the transendothelial and the interendothelial pathway [9, 29]. We therefore define mass transfer from the vasculature to the IF based on the Staverman-Kedem-Katchalsky

equation as

$$\overset{\text{NP}v\to\text{NP}l}{M} = \overset{\text{NP}v\to\text{NP}l}{M_{\text{inter}}} + \overset{\text{NP}v\to\text{NP}l}{M_{\text{trans}}}$$

$$= \underbrace{\rho^v \cdot L_p^v \cdot \frac{S}{V} \left[ p^v - p^l - \sigma \left( \pi^v - \pi^l \right) \right] \Delta\omega_{\text{lm}}^{\text{NP}} \cdot \varepsilon^v}_{\substack{\text{Interendothelial} \\ \text{pathway}}}$$

$$+ \underbrace{\rho^v \cdot P^v \cdot \frac{S}{V} \left\langle \omega^{\text{NP}\bar{v}} - \omega^{\text{NP}\bar{l}} \right\rangle_+ \cdot \varepsilon^v}_{\substack{\text{Transendothelial} \\ \text{pathway}}}$$

(4)

with the log-mean concentration within the pore given by

$$\Delta\omega_{\text{lm}}^{\text{NP}} = \frac{\omega^{\text{NP}\bar{v}} - \omega^{\text{NP}\bar{l}}}{\log\left( \omega^{\text{NP}\bar{v}} / \omega^{\text{NP}\bar{l}} \right)} \simeq \frac{\omega^{\text{NP}\bar{v}} + \omega^{\text{NP}\bar{l}}}{2}$$

(5)

similar to Jain [30]. The first term describes transport through the interendothelial pathway, which is also called intercellular extravasation [9]. This extravasation is a convective process, meaning that the nanoparticles are dragged by the transvascular fluid flow [30]. The endothelial cells around normal capillary vessels are tightly lined so that larger molecules are not able to pass through the space between cells. Due to abnormal vessel characteristics of the tumour vasculature, endothelial cells in this area are poorly aligned leading to gaps between adjacent cells [31]. These gaps can reach 100 nm to 500 nm in size [9]. The leaky and hyperpermeable vessel walls result in fluid extravasation which passively transports nanoparticles from the vasculature through pores or fenestrations into the IF [29]. The mass transfer via the interendothelial pathways is described by a Starling equation with the surface-to-volume ratio $S/V$ and the hydraulic conductivity of the blood-vessel wall given by

$$L_p^v = \frac{\gamma_p \, r_0^2}{8\mu^v t}$$

(6)

with pore radius $r_0$ and vessel wall thickness $t$ as defined by Stylianopoulos and Jain [32]. The fraction of pores $\gamma_p$ defines the fraction of the endothelium surface occupied by pores [33]. In addition, $\sigma(\pi^v - \pi^l)$ describes the oncotic pressure difference between blood vessels and IF.

The second transport mechanism across vessel walls is the transendothelial pathway. Nanoparticles are able to diffuse through the capillary vessel-wall, e.g. through interconnected cytoplasmic vesicles and vacuoles [9]. The diffusive flux through the transendothelial pathway depends on the vascular permeability $P^v$ and the mass fraction difference of nanoparticles across the vessel wall $\Delta\omega^{\text{NP}} = \omega^{\text{NP}\bar{v}} - \omega^{\text{NP}\bar{l}}$. By using Macaulay brackets $\langle \cdot \rangle_+$, we only allow diffusive flux from the vessels to the IF.

In addition to blood vessels, lymph vessels contribute to mass transfer with the IF. In normal tissues, extravasated fluid and molecules are absorbed by the lymph vessels [34]. However, tumours lack a functioning lymphatic system, resulting in inefficient drainage of fluid [35]. The uptake of nanoparticles dispersed in the IF by the lymph system is written as

$$\overset{\text{NP}l\to\text{NP}ly}{M_{\text{drain}}} = \rho^l \cdot \left( L_p \frac{S}{V} \right)^{ly} \cdot \langle p^l - p^{ly} \rangle_+ \cdot \left\langle 1 - \frac{p^t}{p_{\text{coll}}^{ly}} \right\rangle_+ \cdot \omega^{\text{NP}\bar{l}}$$

(7)

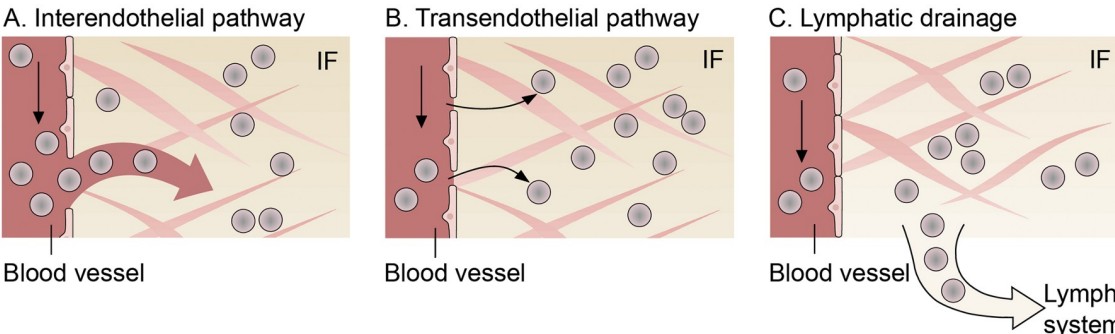

**Fig 2. Mechanisms for nanoparticle transport to and from the interstitial fluid (IF).** Transcapillary exchange of nanoparticles from the vasculature to the IF consisting of (A) the interendothelial and (B) the transendothelial pathway; (C) Lymphatic drainage for transport from the IF to the lymph system.

with the lymphatic filtration coefficient $\left(L_p \frac{S}{V}\right)^{ly}$. This is similar to the mass transfer term for drainage of fluid from the IF as presented in Kremheller et al. [13, 14]. In the following, we assume $p^{ly} \approx 0$. Above the collapsing pressure $p_{\mathrm{coll}}^{ly}$, lymphatic drainage is impaired and no fluid or particles are taken up by the lymph system.

In sum, the mass transfer of nanoparticles to and from the IF comprises three terms

$$\sum_{\kappa \in \mathcal{J}_{cl}} \overset{\mathrm{NP}\kappa \to \mathrm{NP}l}{M} = \underbrace{\overset{\mathrm{NP}v \to \mathrm{NP}l}{M_{\mathrm{inter}}}}_{\substack{\text{Interendothelial}\\ \text{Fig 2A}}} + \underbrace{\overset{\mathrm{NP}v \to \mathrm{NP}l}{M_{\mathrm{trans}}}}_{\substack{\text{Transendothelial}\\ \text{Fig 2B}}} - \underbrace{\overset{\mathrm{NP}l \to \mathrm{NP}ly}{M_{\mathrm{drain}}}}_{\substack{\text{Drainage}\\ \text{Fig 2C}}} \tag{8}$$

where the physical interpretation of each term is depicted in Fig 2. We prescribe that tumour growth does not influence the mass balance of nanoparticles dispersed in the IF and that there is no intra-phase reaction term for nanoparticles, that is, we set in Eq (3)

$$\overset{\mathrm{NP}l \to \mathrm{NP}t}{M} + \varepsilon^l r^{\mathrm{NP}l} - \omega^{\mathrm{NP}\bar{l}}\left(-\overset{l \to t}{M_{\mathrm{growth}}}\right) = 0. \tag{9}$$

## 3 Numerical examples, results and discussion

Initially, we employ our framework, as described in Section 3.1 and S1 Appendix, to study the symmetrical growth of a vascular tumour. We then use this result of a grown tumour and its microenvironment as a starting point for a systematic nanoparticle transport study in Section 3.2. The setup of the geometry is inspired by the experiments conducted by Ziemys et al. [36] and thus is of practical biological and clinical relevance. We study the influence of different parameters which also are of biological interest in order to characterise the transport pheno-type of the tumour. We moreover investigate how those parameters affect the distribution of nanoparticles in the tumour environment.

### 3.1 Vascular tumour growth

We first present a symmetrical example of two-dimensional growth of a vascular tumour which, in its final grown state, later serves as starting point for further simulations. We analyse a domain of 1 mm × 1 mm where, due to the symmetry of the problem, only one quarter is actually simulated (0.5 mm × 0.5 mm). A circular tumour $\Omega^t$ with an initial radius of $r_0 = 25\ \mu m$

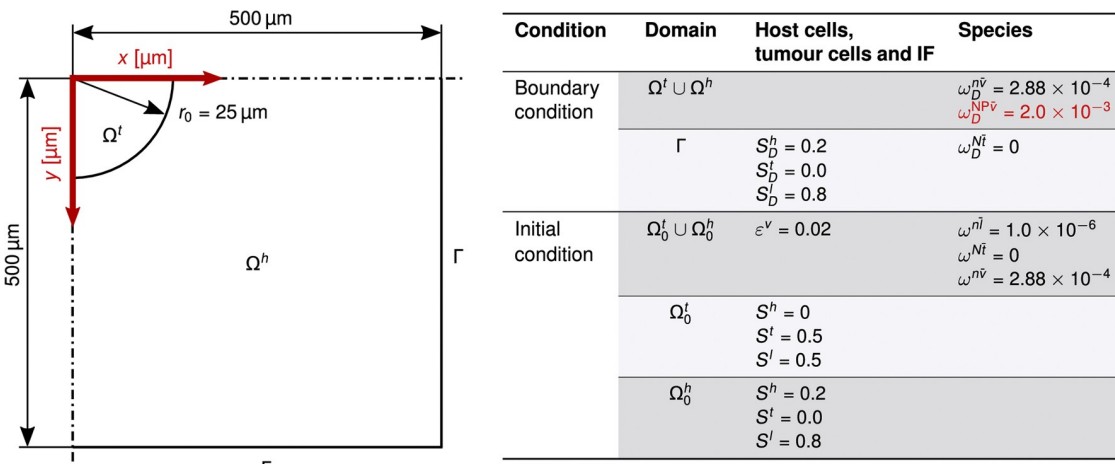

**Fig 3. Geometry and boundary conditions.** Geometry of two-dimensional growth of a vascular tumour with boundary and initial conditions. Note that the initial tumour radius is not sketched to scale. The boundary condition for the study of nanoparticle transport in Section 3.2 is given by the mass fraction of nanoparticles in the vasculature $\omega_D^{\mathrm{NP}\bar{v}}$ in red. The index $D$ marks Dirichlet values.

is growing in host tissue $\Omega^h$ (see Fig 3). This could either be a primary tumour or a metastasis seeded by a parental tumour. The tissue is vascularised with an initial blood vessel volume fraction of $\varepsilon_0^v = 0.02$. This value is estimated from the data of Secomb et al. [37] and in accordance with Jain [38] who states that the vascular space occupies between 1% and 20% in tumours. We further set the mass fraction of oxygen in the vasculature to $\omega_D^{n\bar{v}} = 2.88 \times 10^{-4}$ which corresponds to a oxygen partial pressure in the vasculature of $P_{\mathrm{oxy}}^v \approx 100$ mmHg. Because we do not study transport phenomena in the vasculature here, we assume that $\omega_D^{n\bar{v}}$ is constant and hence apply it as a Dirichlet boundary condition. Oxygen is provided via transcapillary exchange from blood vessels and thereby reaches the IF.

At the beginning, tumour cells have a saturation of $S^t = 0.5$ in the tumour domain $\Omega_0^t$ and $S^t = 0$ in the host domain $\Omega_0^h$. Moreover, we do not have any necrotic tumour cells at the initial state. Those depend on reaction terms of the tumour growth model and form at later time steps. On all other boundaries, we apply a zero Neumann condition with $\frac{\partial B}{\partial \boldsymbol{n}} = 0$ where $B$ denotes the corresponding primary variable at the boundary, e.g. differential pressures, and $\boldsymbol{n}$ the outer unit normal vector.

The domain for each field is discretised with $120 \times 120$ bilinear elements, assuming a plain strain case. The structure, fluid and species transport meshes are conforming. The time step is $\Delta t = 1800$ s and in total 320 time steps are simulated. This describes tumour growth in a time frame of 160 hours ($\approx 6.5$ days). For all fields, a one-step-$\theta$ time integration scheme with $\theta = 1.0$ is applied.

S1 Appendix lists all parameters. Most parameters are based on available literature and have been previously employed [13, 39]. As the constitutive law for the ECM, we employ a Neo-Hookean material law with an initial volume fraction of the ECM of $\varepsilon_0^s = 1 - \varepsilon_0 - \varepsilon_0^v = 0.2$.

**3.1.1 Results.** The volume fraction of living tumour cells $\varepsilon^{\mathrm{LTC}} = \varepsilon(1 - \omega^{N\bar{t}})$ after 160 hours is shown in Fig 4A, where the white contour line indicates the edge of the tumour. The slightly non spherical shape of the tumour results from the boundary conditions being applied to a quadratic domain. After a growth phase of 160 hours, the tumour has reached a size of $r = 440$ μm. Here and in the following, the edge of the tumour is defined as $S^t = 0.1$. Based on this, the radius is estimated as the mean value of the Euclidean distance of the edge to the

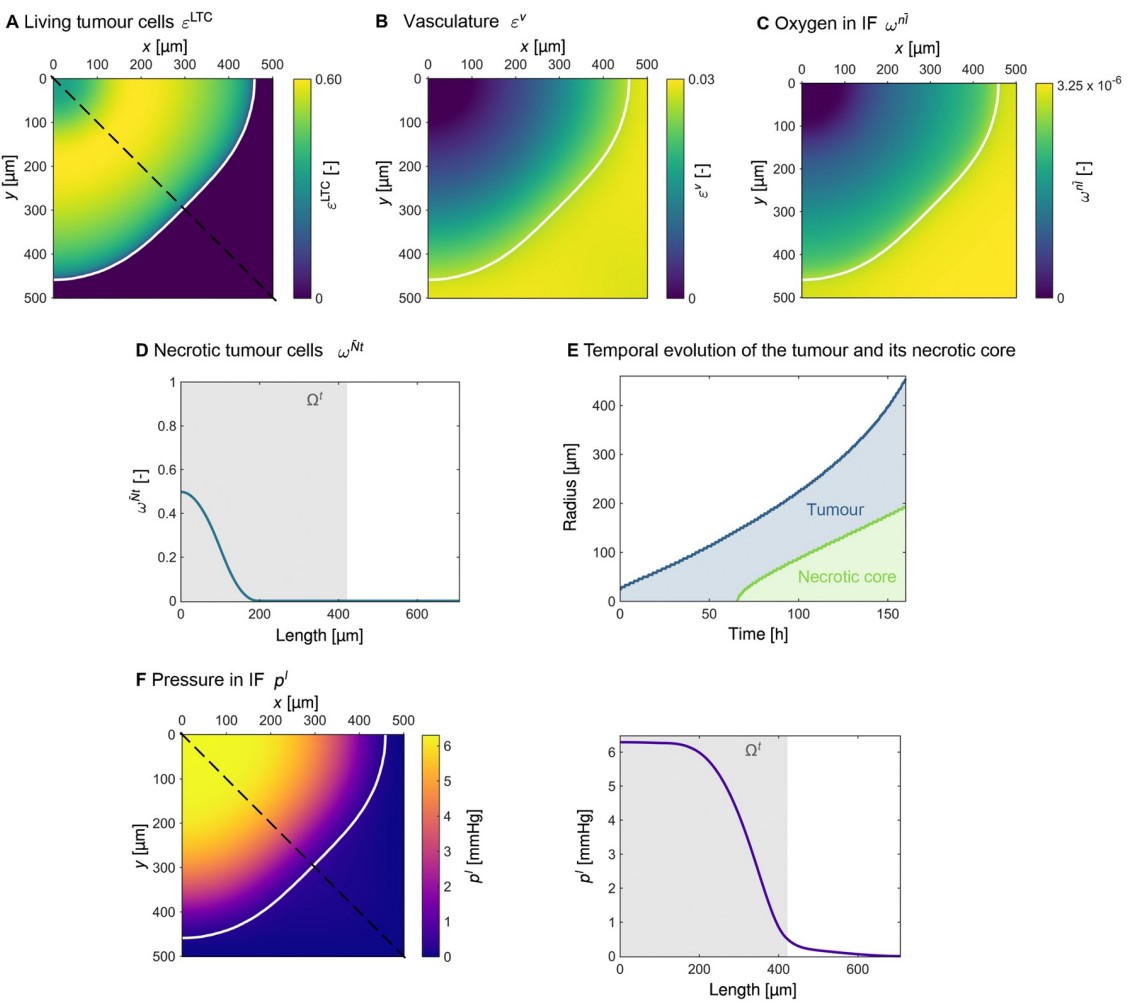

**Fig 4. Results of tumour growth after 120 hours.** The white contour line indicates the edge of the tumour defined as $S^t = 0.1$. (A) Volume fraction of living tumour cells $\varepsilon^{\text{LTC}} = \varepsilon(1 - \omega^{Nt})$. (B) Volume fraction of vasculature $\varepsilon^v$. (C) Mass fraction of oxygen in the interstitial fluid (IF) $\omega^{n\bar{l}}$. (D) Mass fraction of necrotic tumour cells $\omega^{Nt}$ visualised over the cut marked by the dashed line in Fig 4A. (E) Temporal evolution of the radius of the tumour and of its necrotic core. (F) Pressure $p^l$ in the IF.

centre of the tumour (upper left corner of the domain). Fig 4A further shows that the majority of living tumour cells can be found in the tumour periphery, whereas the tumour core consists of necrotic cells. The volume fraction of vasculature depicted in Fig 4B presents the effect of our model for blood-vessel collapse (see S1 Appendix for further details). Considerably fewer blood vessels can be found in the tumour area including an inner core of approximately 100 μm that contains no vessels at all.

The distribution of oxygen in the IF $\omega^{n\bar{l}}$, depicted in Fig 4C, follows a similar pattern. While the oxygen level in the host cells' region around the tumour is $\omega^{n\bar{l}} = 3.25 \times 10^{-6}$, the oxygen level drops significantly when moving towards the central region of the tumour. The innermost part of the tumour is poorly supplied with oxygen. Therefore, tumour cells in this region become necrotic, and a necrotic core evolves. In the centre of the tumour approximately 50% of the tumour cells are necrotic, as can be seen in Fig 4D. A temporal analysis of the size of the tumour and of its necrotic core, as presented in Fig 4E, shows that the necrotic core starts to

develop when the tumour reaches a radius of more than 100 μm, which is the case after 65 hours of growth.

Fig 4F shows the increased interstitial pressure in the tumour. The maximum pressure reached in the IF is $p^l_{max} = 840\,\text{Pa} = 6.3\,\text{mmHg}$. As can be seen in Fig 4F, the IF pressure is high and constant in the central region of the tumour but decreases steeply in the tumour periphery.

**3.1.2 Discussion.** Similar to other tumour models, e.g. Cui et al. [40] or Macklin et al. [41], our results show that the central region of the tumour mainly contains necrotic cells, while the outer shell consists of proliferating cells. This structure of the tumour (necrotic core + viable outer shell) is attributed to the non-uniform nutrient distribution [40]. The tumour cells close to the outer surface receive sufficient oxygen to proliferate. On the contrary, the oxygen level inside the tumour core falls below the critical limit, thereby causing cell death.

According to Carmeliet and Jain [35], cells located more than 100 μm away from the closest capillary become hypoxic because 100 μm is the diffusion limit for oxygen. The development of a necrotic core therefore starts when the tumour reaches a radius of more than 100 μm, as the temporal analysis of tumour growth shows in Fig 4E.

The maximum IF pressure predicted by our model $p^l_{max} = 830\,\text{Pa} = 6.3\,\text{mmHg}$ lies within the range of 5 mmHg to 10 mmHg proposed by Dewhirst et al. [42]. Nevertheless, the raised IF pressure can reach values as high as 60 mmHg [43]. A plateau of IF pressure in the central region of the tumour is typical [43, 44] and has also been observed in *in vivo* experiments by Boucher et al. [45]. As described by Heldin et al. [43], the elevated IF pressure stems from deficiencies in blood and lymph vessel function. Due to blood-vessel leakiness, fluid from the vasculature is transported into the IF, while at the same time lymphatic drainage inside the tumour is impaired. This combination results in fluid accumulation in the tumour region, thereby causing increased IF pressure.

Vessel compression is a hallmark shared by all solid tumours [46]. The growing tumour pushes against its surrounding microenvironment, thereby collapsing both blood and lymphatic vessels. Provenzano et al. [47] conclude from their experimental study that elevated IF pressure induces blood-vessel collapse. By contrast, Chauhan et al. [48] state that IF pressure cannot compress and collapse blood vessels; rather, the compression of blood vessels is caused by solid stresses. Based on the findings of Padera et al. [49], which indicate that proliferating tumour cells cause the collapse of blood vessels, we use $p^t$ as critical value. Compared to the model for blood-vessel collapse employed in Vavourakis et al. [33], we use a heuristic approach where we set the parameters in such a way that blood-vessel collapse is restricted to the tumour region. This simple approach is used because the main focus of this study is not on blood-vessel collapse but rather on the distribution of nanoparticles. We therefore avoid the latter being influenced by a complex model for blood-vessel collapse. Notwithstanding, we are aware of the fact that more complex models are necessary to further investigate the exact reasons for blood-vessel collapse.

To sum up, after 160 hours of growth, the tumour has developed a non-perfused core with collapsed blood vessels as well as increased interstitial pressure.

## 3.2 Nanoparticle transport study

The pathway nanoparticles take from an intravenous infusion site to the tumour comprises three main phases [42]: Transport in the vasculature with subsequent crossing of vessel walls; transport in the IF; and finally removal by the lymph system. These three transport phases are shown schematically in Fig 5. The transport of nanoparticles along the described pathway is affected by several biophysical barriers [50]. We therefore investigate the distribution of

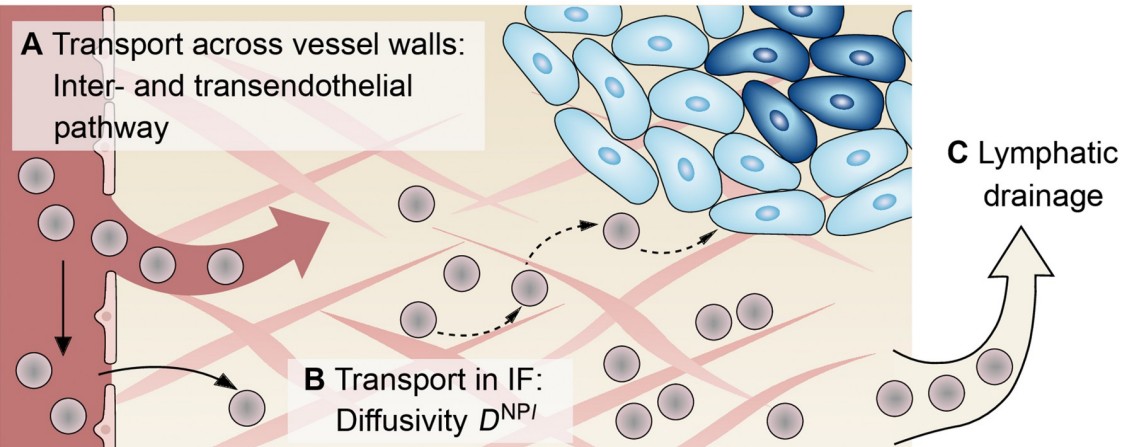

**Fig 5. Transport barriers to nanoparticle transport.** (A) Transport across the vessel wall via the interendothelial and/or the transendothelial pathway. (B) Transport in the IF characterised by the diffusivity $D^{\mathrm{NP}l}$. (C) Removal by the lymph system via lymphatic drainage.

nanoparticles in the IF $\omega^{\mathrm{NP}\bar{l}}$ based on a systematic variation of the parameters which characterise the three transport phases.

In order to simulate transport and accumulation of nanoparticles within the tumour, we use the final result after 160 hours of tumour growth (see Fig 4) as a starting point for the nanoparticle simulations. At this point, the grown tumour displays common characteristics of solid tumours, as described in Section 3.1, which complicate efficient nanoparticle delivery.

To study nanoparticle distributions, we simulate a treatment with an intravenous infusion of the particles. We assume that an intravenous infusion of nanoparticles directly influences their mass fraction in the blood in the entire systemic circulation [42]. Moreover, the administration of a drug by intravenous infusion reaches a steady state where the drug concentration in the blood is constant [51]. We therefore prescribe a constant value $\omega_D^{\mathrm{NP}\bar{v}} = 2.0 \times 10^{-3}$ on the entire domain $\Omega^t \cup \Omega^h$ to account for administered nanoparticles in the vasculature (see Fig 3). This value is of the same order of magnitude as the nanoparticle reference concentration of Nabil et al. [23]. Moreover, this value for the mass fraction of nanoparticles is small enough not to influence the global physical properties of the solvent, in this case blood [15]. Terentyuk et al. [52] use a 20 min injection period for experiments where a nanoshell suspension is tested for tumour treatment in rats. Similarly, Nabil et al. [23] start with an initial injection period of 20 min for numerical experiments with nanoparticles. Based on these examples, we analyse a time interval of 20 min with a time step of $\Delta t = 60$ s for all following simulations. The tumour growth equations, as described in S1 Appendix, are still evaluated, but due to the different time scales of tumour growth and nanoparticle transport the tumour is effectively stationary in the configuration depicted in Fig 4.

**3.2.1 Transport across vessel walls. 3.2.1.1 Results.** In the following section, we analyse the influence of different transvascular pathways on the distribution of nanoparticles in the IF. We start with a constant vascular permeability of $P^v = 3.5 \times 10^{-4}$ mm/s [53] and vary the amount of convective transport across the vessel walls. Based on Eq (6), we now consider different pore sizes, namely $r_0 \in \{50$ nm, 100 nm, 150 nm, 200 nm$\}$ which is within the range proposed by Stylianopoulos and Jain [32]. We set $L_p^{\mathrm{NP}ly} = 0$ in order to only analyse transport across vessel walls.

Fig 6A1 visualises that the nanoparticles mainly accumulate in the region with a high volume fraction of vasculature $\varepsilon^v$ which in our case coincides with the region of host cells around the tumour which is still well perfused (see Fig 4B and 4C). The mass fraction of nanoparticles in the IF $\omega^{\mathrm{NP}\bar{l}}$ rises from $1.0 \times 10^{-3}$ to $1.7 \times 10^{-3}$ for a pore radius of $r_0 = 50$ nm and $r_0 = 200$ nm, respectively. In contrast, considerably fewer particles reach the tumour core in all four cases, on average $0.3 \times 10^{-3}$ in the tumour centre.

We further investigate the contribution of the transcellular pathway where particles are transported across endothelial cells either trough a series of linked vesicles or through a single vesicle [9]. We use a constant pore radius of $r_0 = 150$ nm and vary the vascular permeability $P^v$. We choose dextran as model carrier to analyse the effect of different vascular permeabilities on the distribution of nanoparticles in the IF. In Fig 7, we summarise permeability coefficients of dextrans with different molecular weights, as determined by Dreher et al. [54], Chou et al. [55] and Ho et al. [53]. The summary in Fig 7 reveals that the measured permeability coefficients differ by more than one order of magnitude. As a baseline, we use the purely convective transport via the interendothelial pathway and set $P^v = 0$. We then choose the values for 10 kDa to exemplarily study the influence on the distribution of nanoparticles in the IF.

The results are depicted in Fig 6A2. Compared to Fig 6A1, the qualitative distribution of nanoparticles in the IF does not change under the influence of transcellular transport. However, the absolute mass fraction of nanoparticles in the IF changes. In our example with a fixed pore radius of $r_0 = 150$ nm, a permeability coefficient of $P^v = 3.2 \times 10^{-5}$ mm/s as determined by Dreher et al. [54] has little influence on the nanoparticles distribution, which rises by only 5% when compared to the baseline simulation with $P^v = 0$. In contrast, the values $P^v = 3.5 \times 10^{-4}$ mm/s by Ho et al. [53] and $P^v = 1.28 \times 10^{-3}$ mm/s by Chou et al. [55] result in an increase of the mass fraction of nanoparticles in the IF by 41% and 84%, respectively.

**3.2.1.2 Discussion**. The variation of pore radius $r_0$ and vessel wall permeability $P^v$ shows that particles mainly accumulate in the region outside the tumour in both cases. This distribution of nanoparticles is due to the fact that the convective mass transfer $M_{\mathrm{inter}}$ and the diffusive mass transfer $M_{\mathrm{trans}}$ of nanoparticles across the vessel walls as prescribed by Eq (4) is biggest in the region outside the tumour which is still well perfused. However, to reach the core of the tumour, diffusion inside the IF is necessary, which is the inhibiting factor here.

As depicted in Fig 7, the permeability coefficients measured by Dreher et al. [54], Chou et al. [55] and Ho et al. [53] differ by more than one order of magnitude. Dreher et al. [54] use a mouse model to measure the apparent permeability *in vivo*. This includes an unknown influence of convection as stated by the authors. Chou et al. [55] use the Kedem-Katchalsky equation to fit permeability and diffusivity of dextrans to the data of Dreher et al. [54]. Ho et al. [53], in contrast, reason that a systematic quantitative study of permeabilities utilising animal models is difficult. The authors therefore propose an *in vitro* model based on microfluidics. This model allows a quantitative study of the extravasation, as well as control over the pathway taken by the nanoparticles. We therefore use the permeability coefficient determined by Ho et al. [53] as the default value (see Table 1).

Our findings are consistent with the fact that the extravasation of nanoparticles through both pathways (trans- and interendothelial route) influence the amount of nanoparticles reaching the tumour. The remaining question however is which mechanism is dominant. Wilhelm et al. [9] summarise that distinguishing between transvascular transport through intercellular gaps or transendothelial cell pores is difficult and that intercellular gaps have not yet been definitively established. Nevertheless, nanotechnology has focused on transport through intercellular gaps as described by the *enhanced permeability and retention* (EPR) effect, and so far has not overcome major problems with poor delivery efficiency [9, 57, 58]. We therefore

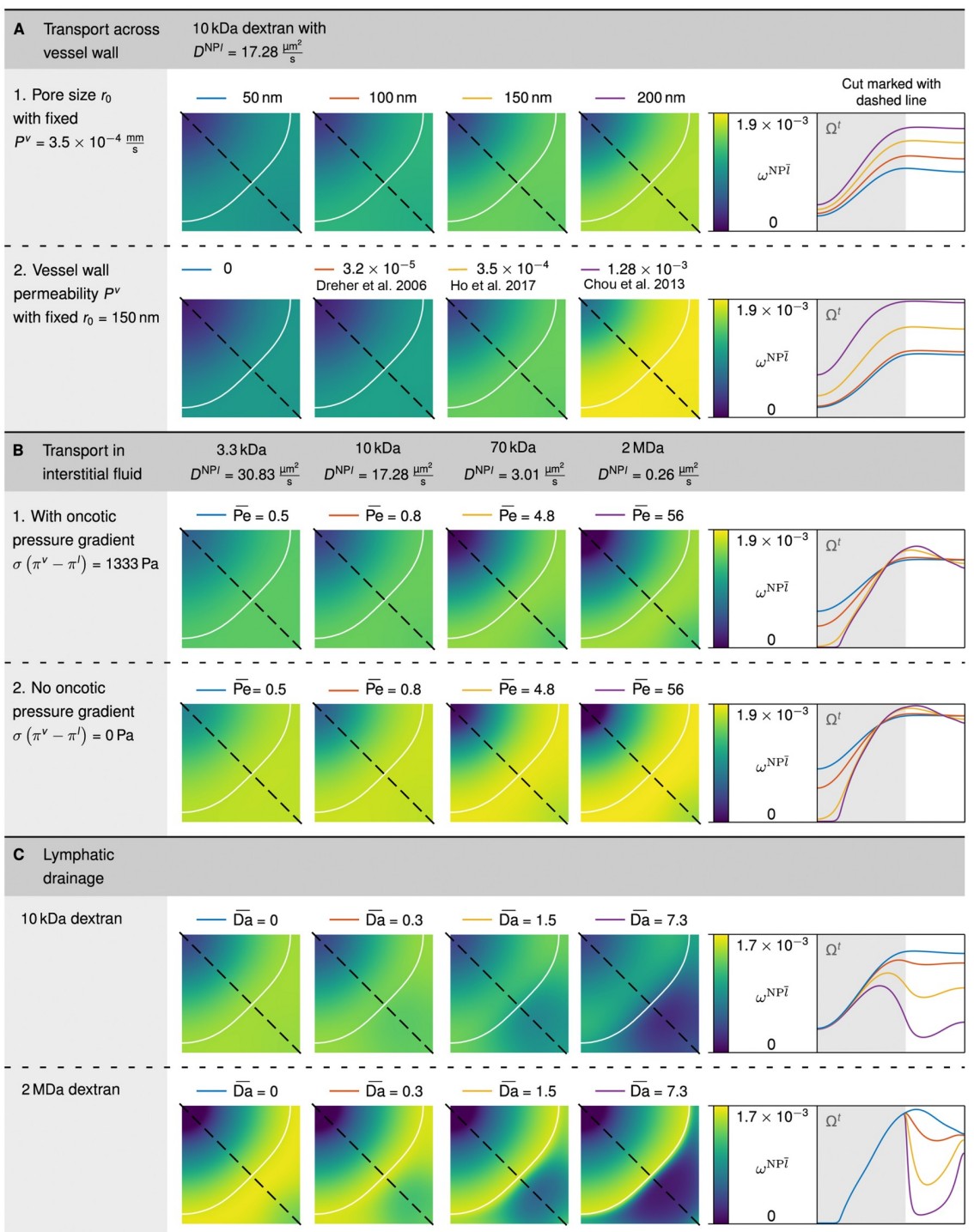

**Fig 6. Nanoparticle distribution in the interstitial fluid $\omega^{\mathrm{NP}l}$ after a 20 min injection period.** (A) Influence of transport across vessel walls via the inter- and the transendothelial pathway. (B) Influence of transport characteristics in the interstitial fluid with different Péclet numbers. (C) Influence of lymphatic drainage with different Damköhler numbers.

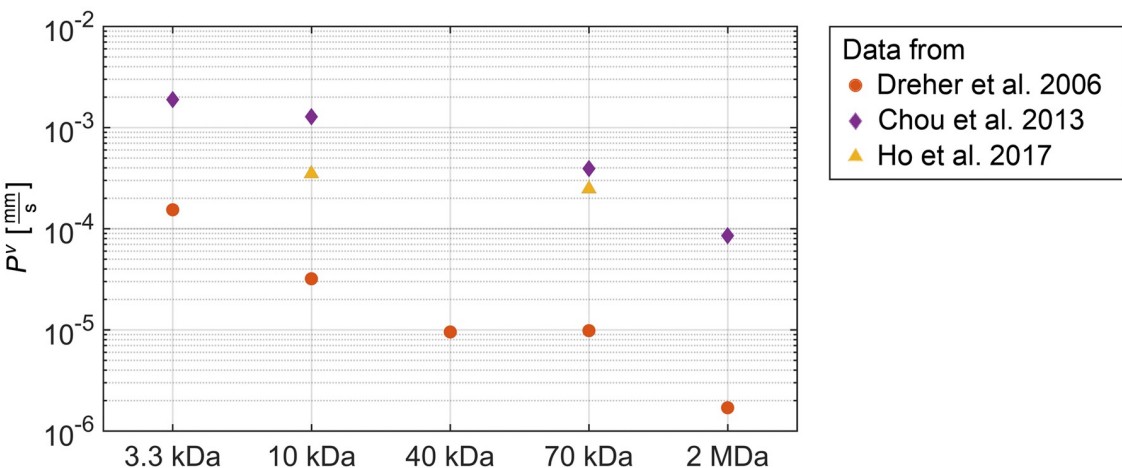

**Fig 7. Values for the vascular permeability coefficient.** Vascular permeability coefficient $P^v$ of dextrans with a molecular weight of 3.3 kDa, 10 kDa, 40 kDa, 70 kDa and 2 MDa summarised from data by Dreher et al. [54], Chou et al. [55] and Ho et al. [53].

use a combination of trans- and interendothelial routes, where both pathways contribute a considerable amount of nanoparticle transport to the IF by setting the pore radius to $r_0 = 150$ nm and the vascular permeability coefficient to $P^v = 3.5 \times 10^{-4}$ mm/s [53] for all following simulations. Jain and Stylianopoulos [31] nonetheless state that tumour vasculature is not always as leaky as postulated in the description of the EPR effect and thus Wilhelm et al. [9] conclude that future research should concentrate on the transendothelium transport mechanism. Here, we have shown that the transendothelial pathway indeed plays a major role and can increase the efficiency of drug delivery.

**3.2.2 Transport in the IF. 3.2.2.1 Results.** We further investigate the transport of nanoparticles in the IF which is driven by diffusion and convection. To study the transport in the IF, we define the Péclet number as the ratio of convective transport rate to diffusive transport rate in the IF

$$\text{Pe} = \frac{u^l \cdot L}{\varepsilon S^l D^{\text{NP}l}} = \frac{(S^l)^{A_l - 1} k \cdot L}{\varepsilon \mu^l D^{\text{NP}l}} \nabla p^l \tag{10}$$

where we use the Darcy equation for the convective velocity $u^l$. The characteristic length is set to $L = (A_{\text{tumour}}/P_{\text{tumour}}) = 220$ μm with $A_{\text{tumour}}$ being the surface of the tumour and $P_{\text{tumour}}$ the perimeter. The pressure gradient $\nabla p^l$ is calculated using central differences. For the further analysis, we use the mean value $\overline{\text{Pe}}$ of the Péclet number on the entire domain $\Omega^t \cup \Omega^h$. We

**Table 1. Parameters for nanoparticle transport across vessel walls.**

| Symbol | Name | Value | Unit | Source |
|---|---|---|---|---|
| $\gamma_p$ | Fraction of pores | $10 \times 10^{-4}$ | - | [33] |
| $\mu^v$ | Viscosity of blood | $4 \times 10^{-3}$ | Pa s | [32] |
| $t$ | Thickness of vessel wall | 1 | μm | [56] |
| $S/V$ | Surface-to-volume ratio | 20 | mm$^{-1}$ | [44] |
| $r_0$ | Pore radius | 50, 100, **150** $^{(\star)}$, 200 | nm | [32] |
| $P^v$ | Blood-vessel wall permeability of 10 kDa dextran | 0, $3.2 \times 10^{-5}$, $\mathbf{3.5 \times 10^{-4}}$ $^{(\star)}$, $1.28 \times 10^{-3}$ | mm/s | [53–55] |

$^{(\star)}$ The bold value marks the default value used for the nanoparticle transport study in Section 3.2.

**Table 2. Characteristics of dextrans of various molecular weight.** Values are based on Chou et al. [55].

| Dextran | Diffusivity | Péclet number | Size[(*)] |
|---|---|---|---|
| Molecular weight (kDa) | $D^{\mathrm{NP}l}$ ($\mu m^2$/s) | $\overline{\mathrm{Pe}}$ (−) | (nm) |
| 3.3 | 30.83 | 0.5 | 1.6 |
| 10 | 17.28 | 0.8 | 2.8 |
| 70 | 3.01 | 4.8 | 7.3 |
| 2000 (= 2MDa) | 0.26 | 56 | 25 |

[(*)] Molecular size is expressed as the hydrodynamic radius determined by the Stokes-Einstein equation. Values are taken from Dreher et al. [54].

now include both inter- and transendothelium pathways for transport across vessel walls. Here, we use a constant permeability coefficient of $P^v = 3.5 \times 10^{-4}$ mm/s [53] and a pore radius of $r_0 = 150$ nm. In order to investigate nanoparticle distributions in different transport regimes, we now compare dextrans with different molecular weights, similar to the experiments performed by Ziemys et al. [36]. The interstitial diffusivities for 3.3 kDa, 10 kDa, 70 kDa and 2 MDa dextran determined by Chou et al. [55] result in Péclet numbers in the range of 0.5 to 56 (see Table 2). This allows us to compare diffusion-dominated with convection-dominated transport in the IF. The convective transport rate is influenced by the pressure gradient $\nabla p^l$. The diffusive transport rate depends on the particle size but at same time on the ECM which, due to its dense structure, impedes diffusive transport [31, 50]. We use the Péclet number to analyse the influence of all these factors collectively.

Fig 6B1 presents the distribution of dextrans with various molecular weights in the IF. In the diffusion-dominated case, where $\overline{\mathrm{Pe}} = 0.5$ and $D^{\mathrm{NP}l} = 30.83$ $\mu m^2$/s for 3.3 kDa, the nanoparticles spread more uniformly across the domain. In contrast, in the convection-dominated case, where $\overline{\mathrm{Pe}} = 56$ and $D^{\mathrm{NP}l} = 0.26$ $\mu m^2$/s, the particles accumulate at the tumour edge. In particular, no particles reach the core of the tumour, which coincides with the region without blood vessels (see Fig 4B). We thus observe that the nanoparticles can only reach the whole tumour domain if the flow in the IF is dominated by diffusion. If convection dominates the flow characteristics, particles do not reach the core of the tumour or any regions located further away from functioning blood vessels.

For all simulations so far, we included an oncotic pressure difference of $\sigma(\pi^v - \pi^l) = 1333$ Pa, based on Wu et al. [59] and Baxter and Jain [60]. This implies two things. First, the oncotic pressure in the tumour interstitium $\pi^l$ differs significantly from the oncotic pressure in blood plasma $\pi^v$ and hence, an oncotic pressure gradient is present. Second, the reflection coefficient $\sigma$, which influences the effectiveness of the pressure gradient, must be greater than zero. In contrast to the findings of Baxter and Jain [60], Stroher et al. [61] state that, based on their measurements, the oncotic pressure gradient across the blood-vessel wall is low. Moreover, Tong et al. [62] assume a reflection coefficient close to zero because of large pores in the vessel walls and the consequent leakiness. Hence, the question arises as to whether and how the presence of an oncotic pressure difference across the tumour microvascular wall influences the distribution of nanoparticles. We therefore analyse the distributions of dextran in the extreme case of $\sigma(\pi^v - \pi^l) = 0$ Pa. The results in Fig 6B2 show that the qualitative distribution of particles in the IF is similar to the results with an oncotic pressure gradient (compare Fig 6B1). However, a quantitative comparison reveals that approximately 20% more particles reach the IF when no significant oncotic pressure gradient is present.

**3.2.2.2 Discussion.** After crossing the vascular barrier, nanoparticles have to navigate through the tumour microenvironment in order to reach cancerous cells. The IF pressure in

the tumour can be 10 to 40 times higher than in host tissue thereby creating an outward pressure gradient [43]. This increases the outward interstitial flow and thereby limits convective transport into the tumour region [9]. Nanoparticle transport to the centre of the tumour is thus impeded and nanoparticles hardly reach the tumour core or any regions located further away from functioning vasculature as shown in Fig 6B. As summed up by Zhang et al. [63], the elevated IF pressure is one of the main factors that hinders effective tumour penetration of nanoparticles. Goel et al. [64] state that vascular normalisation reduces tumour hypoxia as well as the IF pressure. Based on those findings, Chauhan et al. [65] suggest that vascular normalisation improves the convective delivery of nanoparticles.

Ziemys et al. [36] conclude from their results that the dominant transport mechanism is diffusion. Our results show that only if diffusive transport in the IF is significant, do nanoparticles penetrate into the centre of the tumour [66]. However, Nichols and Bae [57] as well as Danhier and Preat [67] state that the higher and heterogeneous density of human ECM leads to regions that are inaccessible to nanoparticles. In consequence, the IF pressure gradient impedes convective transport, while the denser structure of the tumour ECM limits diffusive transport [31].

In sum, the movement of nanoparticles in the IF is considerably limited, and particles remain in proximity to where they cross the vascular barrier. Consequently, nanoparticles only reach well vascularised regions, and the highest concentrations are located near the vascular surface [54]. Since tumour cores are poorly perfused [68], nanoparticles and anti-cancer agents transported by nanoparticles do not reach those regions. Our model has proven to be well suited to capture this behaviour.

**3.2.3 Influence of lymphatic drainage.** **3.2.3.1 Results**. Now, we include lymphatic drainage of IF and of the nanoparticles contained therein. The lymphatic system generally drains excessive fluid and thereby removes waste products and foreign substances [69]. To analyse its influence on the distribution of nanoparticles in the tumour, we assume that the lymphatic system recognises the nanoparticles as foreign bodies and thus removes them at an increased rate. Similar to the Péclet number, we define the Damköhler number as the ratio of reactive timescale to convective timescale, as proposed by Shipley and Chapman [70]

$$\mathrm{Da} = \frac{\lambda L}{u^l} \qquad (11)$$

where $\lambda$ represents the rate of species loss in the IF due to lymphatic drainage given by

$$\lambda = \frac{\overset{\mathrm{NP}l \to \mathrm{NP}ly}{M_{\mathrm{drain}}}}{\rho^l} = \left( L_p \frac{S}{V} \right)^{ly} \cdot \langle p^l - p^{ly} \rangle_+ \cdot \left\langle 1 - \frac{p^t}{p^{ly}_{\mathrm{coll}}} \right\rangle_+ \qquad (12)$$

and the convective velocity $u^l$, as also employed in Eq (10). We analyse the resulting distribution of nanoparticles in the IF for different Damköhler numbers. To that end, we vary the hydraulic conductivity of lymph vessels. Baxter and Jain [66] use a lymphatic filtration coefficient $\left( L_p \cdot \frac{S}{V} \right)^{ly} = 1.04 \times 10^{-6} \ (\mathrm{Pa \, s})^{-1}$ for the lymph vessels. To investigate different transport regimes, we employ different values for the lymphatic filtration coefficient in $\overset{\mathrm{NP}l \to \mathrm{NP}ly}{M_{\mathrm{drain}}}$, resulting in a mean value of the Damköhler number of $\overline{\mathrm{Da}} = 0.3, 1.5$ and $7.3$ on the entire domain. We again use a constant permeability coefficient of $P^v = 3.5 \times 10^{-4}$ mm/s [53] and a pore radius of $r_0 = 150$ nm.

Above the collapsing pressure $p^{ly}_{\mathrm{coll}}$, the interstitial hypertension in the tumour causes lymph vessels to collapse, and thus no fluid is drained [13]. This is the case in the whole tumour zone,

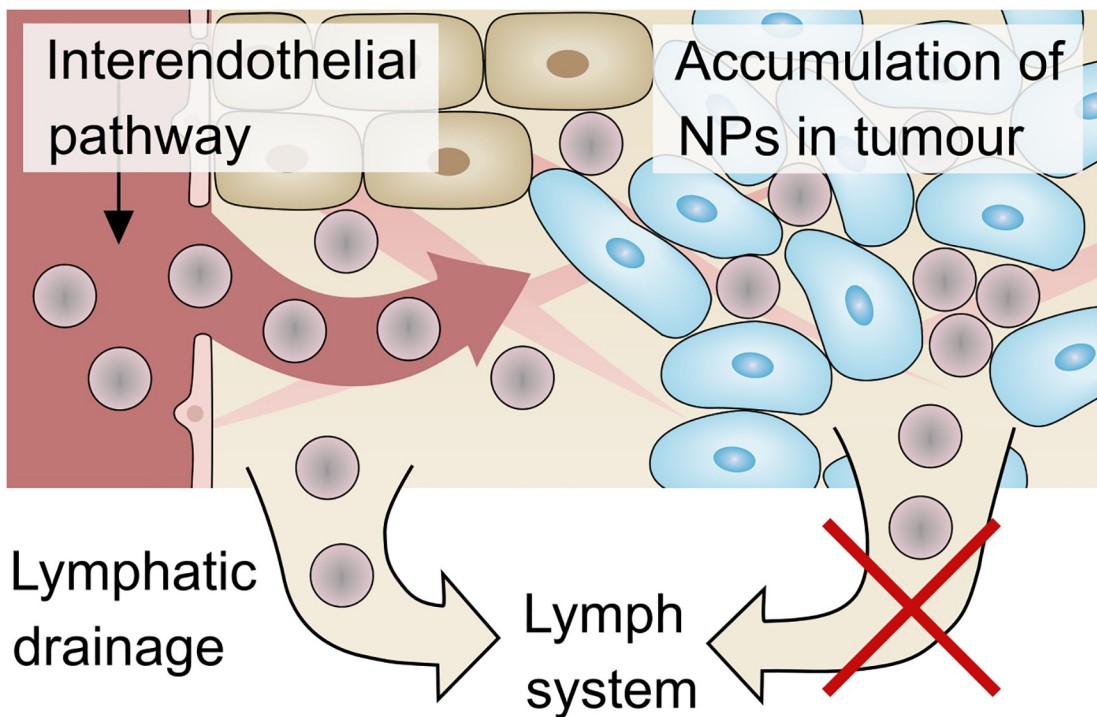

**Fig 8. EPR effect.** Nanoparticles (NPs) leak from the vasculature to the IF. In the tumour region, the lymphatic drainage is impaired and hence the particles are not removed by the lymph system. This results in a passive accumulation of nanoparticles in the tumour. This phenomenon is called the EPR (enhanced permeability and retention) effect [71].

and hence lymphatic drainage is impaired. The drainage of the lymph system therefore only influences regions outside the tumour. The amount of drained fluid increases with increasing hydraulic conductivity, as shown in Fig 6C. In particular, in the case of 2 MDa dextran, where transport in the IF is convection-dominated, the particles clearly accumulate at the outer edge of the tumour.

**3.2.3.2 Discussion**. This interplay of transvascular transport and lymphatic drainage results in an accumulation of nanoparticles in the tumour—a phenomenon called *enhanced permeability and retention* (EPR) effect. Endothelial cells in the vasculature are poorly aligned, leading to large fenestrations. This effect causes extensive leakage of blood plasma components, including macromolecules and nanoparticles, into the interstitium. At the same time, lymphatic clearance is inhibited by the increased interstitial pressure of the tumour. This combination of vascular leakage and impaired lymphatic drainage leads to the passive retention of nanoparticles in the tumour. The EPR effect, as depicted in Fig 8, was first described by Matsumura and Maeda [71] in 1986. It became a gold standard for the design of anti-cancer agents based on macromolecules and nanoparticles with the aim of selectively targeting tumour sites [72].

Since the first mention of the EPR effect more than 30 years ago, the number of publications citing it has increased exponentially, according to Nichols and Bae [57]. Nichols and Bae [57] further state that despite good documentation and the experimental validation of the EPR effect in small animal models, in particular with nanoparticles, the clinical translation of nanomedicine is still limited. Summarising literature from the past ten years, Wilhelm et al. [9] reveal that only 0.7% of administered nanoparticles reach solid tumours in mouse models. In addition, human tumours differ from murine tumours in several major characteristics. The

fact that rodent tumours grow much faster than human tumours leads to several differences which are reviewed in Danhier and Preat [67]. In some human tumours, blood vessels are less leaky than expected, and hence limit extravasation and delivery of nanoparticles [31]. Heterogeneous blood flow in tumours and resulting hypoxic areas hinder the delivery of nanoparticles and drugs to these tumour zones [73].

The nanoparticle distributions in Fig 6C show that the EPR effect can be captured by our model. Nevertheless, proper calibration of the model is important to decide which transport regimes occur *in vivo*. In particular, better experimental characterisation of lymphatic drainage is important in order to validate the mechanisms contributing to the EPR effect. Our results show that the EPR effect might indeed reduce the accumulation of nanoparticles or drugs in host tissue and hence prevent the related side effects. At the same time, it does not improve delivery to the tumour, in particular not to the tumour centre.

## 4 Conclusion

We have extended our vascular multiphase tumour growth model presented in Kremheller et al. [13, 14] to include a suitable model for nanoparticle transport. We have analysed and discussed the resulting nanoparticle distributions in the tumour, in consideration of transport barriers proposed by transport oncophysics [11]. Medical and clinical studies in this context have revealed that several anatomical and physiological factors hinder sufficient penetration of nanoparticles to reach the whole tumour zone. Our tumour model reproduces those transport characteristics, including the high interstitial pressure and the EPR effect. We have shown that the accumulation of nanoparticles in the tumour depends on several parameters, and that nanoparticles do not reach the entire tumour in all cases.

Our model predicts that the amount of nanoparticles reaching the tumour can be increased by considering the transendothelial pathway. Up until now, nanotechnology has focused on transport through the interendothelial pathway, and has not overcome major delivery problems. However, a comprehensive characterisation of the transendothelial pathway lacks precise parameters from a biological point of view, e.g. for vascular permeability. We further deduce from our model predictions that if (or even 'only if') transport in the IF is diffusion-dominated, nanoparticles are able to reach the tumour core. However, diffusion is limited to about 100 μm [42] and the structure of the ECM impairs diffusion. If, on the contrary, the transport is convection-dominated, the high interstitial pressure counteracts transport into the tumour. As a result, nanoparticles remain in proximity to where they cross the vascular border. Since tumour cores are poorly perfused, nanoparticles cannot reach the central regions of the tumour. One possible treatment strategy to overcome these difficulties is vascular normalisation, coined by Jain et al. [74], to restore functionality of the tumour vasculature and thereby improve drug delivery. Only some minor modifications of the model are necessary to include vascular normalisation therapy, which will be a topic of further research. Here, we have used a simple model to consider the collapse of blood vessels in tumours. In future work, we plan to extend our transport study to cover transport in the vasculature including the temporal evolution of the capillary network through angiogenesis and blood-vessel collapse.

In this contribution, the full vascular tumour growth model has served mainly as a means to generate a physically plausible configuration for our nanoparticle transport study. In future work, however, it could have several benefits to study drug delivery in combination with a tumour growth model. This requires the introduction of drug uptake and of a killing mechanism. For instance, the effect of periodic treatments with intermittent tumour growth can be assessed or novel therapies, such as vascular normalisation followed by treatment with a conventional drug, can be integrated more easily.

Many studies assume homogeneous tumours and also a uniform distribution of the nano-particles [75, 76]. In contrast, we model and observe highly uneven distributions which, in a similar form, have been observed *in vivo* as summarised by Wilhelm et al. [9] and limit the clinical translation of nanomedicine. Our model in particular captures the high interstitial pressure in tumours, which hinders nanoparticles from reaching all regions of the tumour. This can give valuable insight into the transport characteristics and accumulation of nanoparticles in tumours, which can be transferred to medical studies.

A notable challenge on the way towards clinical application of tumour simulations as a prognostic tool is experimental validation and verification. Most studies are exclusively *in silico* and include many biological and physical parameters, which often are purely numerical. An advantage of our approach in this context is its physics-based nature, i.e. parameters are real biological or physical quantities. But even though we base our values on literature, extensive further experimental validation is essential and should include *in vitro* and *in vivo* studies as well as clinical analysis. The focus of further research will be the further verification and valida-tion of our model and the employed parameters based on experimental data, as for instance provided by Ziemys et al. [36] and similar studies.

Additionally, tumours are as diverse in their characteristics as patients are [58], and reside in a complex environment which is yet to be fully understood [77]. Model development and experimental validation and verification should therefore complement each other in a feed-back cycle. Computational tumour models in combination with clinical image analysis and data processing should aim at more personalised treatment strategies in order to improve the therapeutic outcome and to limit side-effects. Our approach is not based purely on 'big data' or 'artificial intelligence' but on the underlying physical mechanisms. We believe that only the integration of mechanism-based computational models with statistical data can predict the patient-specific evolution of tumours and the efficacy of treatment strategies [78]. Our vascular multiphase tumour growth model is physics-based and therefore can provide crucial insight into mechanisms that cannot be conclusively investigated either *in vivo* or *in vitro*.

## Supporting information

**S1 Appendix. Nomenclature, additional model equations and parameters.**
(PDF)

## Author Contributions

**Conceptualization:** Bernhard A. Schrefler, Wolfgang A. Wall.

**Formal analysis:** Barbara Wirthl, Johannes Kremheller.

**Funding acquisition:** Bernhard A. Schrefler, Wolfgang A. Wall.

**Investigation:** Barbara Wirthl, Johannes Kremheller.

**Methodology:** Barbara Wirthl, Johannes Kremheller, Bernhard A. Schrefler, Wolfgang A. Wall.

**Project administration:** Bernhard A. Schrefler, Wolfgang A. Wall.

**Resources:** Wolfgang A. Wall.

**Software:** Barbara Wirthl, Johannes Kremheller.

**Supervision:** Bernhard A. Schrefler, Wolfgang A. Wall.

**Visualization:** Barbara Wirthl, Johannes Kremheller.

**Writing – original draft:** Barbara Wirthl, Johannes Kremheller.

**Writing – review & editing:** Barbara Wirthl, Johannes Kremheller, Bernhard A. Schrefler, Wolfgang A. Wall.

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
