## [Decision Letter · Decision Letter 0]

18 Nov 2019

PONE-D-19-26982

Extension of a multiphase tumour growth model to study nanoparticle delivery to solid tumours

PLOS ONE

Dear Dr. Kremheller,

Thank you for submitting your manuscript to PLOS ONE. After careful consideration, we feel that it has merit but does not fully meet PLOS ONE’s publication criteria as it currently stands. Therefore, we invite you to submit a revised version of the manuscript that addresses the points raised during the review process.

The reviewers have suggested/required revisions that are minor in nature. I am in agreement, and so recommend that the authors make the minor revisions before the paper can be accepted.

We would appreciate receiving your revised manuscript by Jan 02 2020 11:59PM. To enhance the reproducibility of your results, we recommend that if applicable you deposit your laboratory protocols in protocols.io, where a protocol can be assigned its own identifier (DOI) such that it can be cited independently in the future. For instructions see: http://journals.plos.org/plosone/s/submission-guidelines#loc-laboratory-protocols

We look forward to receiving your revised manuscript.

Kind regards,

Krishna Garikipati, PhD

Academic Editor

PLOS ONE

Journal Requirements:

Additional Editor Comments:

The reviewers have suggested/required revisions that are minor in nature. I am in agreement, and so am recommending that the authors should make the minor revisions before the paper can be accepted.

Reviewers' comments:

Reviewer's Responses to Questions

**Comments to the Author**

1. Is the manuscript technically sound, and do the data support the conclusions?

Reviewer #1: Yes

Reviewer #2: Yes

2. Has the statistical analysis been performed appropriately and rigorously? 

Reviewer #1: N/A

Reviewer #2: N/A

3. Have the authors made all data underlying the findings in their manuscript fully available?

Reviewer #1: Yes

Reviewer #2: Yes

4. Is the manuscript presented in an intelligible fashion and written in standard English?

Reviewer #1: Yes

Reviewer #2: Yes

5. Review Comments to the Author

Reviewer #1: The authors have previously developed a multiphase model for tumor growth. Here they extend the model to examine the factors influencing nanoparticle delivery to tumors. The main conclusion is that only very small nanoparticles, for which transport is diffusion dominated, can reach the entire tumor. The effects of varying key parameters describing wall permeability to nanoparticles are investigated. The analysis is carefully done and clearly presented, and the results appear plausible. The approach used is more elaborate than would be needed to establish these results. For example, the model is used to simulate the dynamics of tumor growth, but the nanoparticle transport simulations are done for the configuration at 160 hours, which could perhaps have been established based on previously published observations and/or simulations. However, this does not invalidate the results obtained. The strengths and limitations of the approach are appropriately discussed. Overall, while the work does not lead to striking new insights, it is a valid contribution to the literature on this topic, providing quantitative information about the factors influencing nanoparticle delivery to tumors.

Reviewer #2: Overall, I found this paper interesting, especially the nanoparticle transport study. I think the authors' systematic study of the different transport barriers and the sensitivity of the drug transport to different parameters is well designed and satisfies the Plos One criteria for publication. There are a few points that I would like the authors to address and/or discuss in their manuscript.

1. First, the storyline of the paper is built around the extension of a multiphase tumor growth model to study nanoparticle delivery, but the role of the tumor model is minor. All the results and conclusions are focused on the nanoparticle transport. As far as I understand the results in Fig. 6 where produced only with the nanoparticle transport compartment of the model (the tumor dynamics is frozen, right?). If all of these assumptions are correct, I think there is a lot of information in the paper that deviates the reader from the main message, which comes across more weakly.

2. There is a lot of notation in the paper. It is difficult to keep track of the symbols. A table with all the notation would help.

3. In the discussion of Fig. 4, how is the tumor radius defined? (the tumor is not circular). Also how is 'the dege of the tumor' defined? As far as I can see all variables are continuous across the tumor boundary.

4. The images look blurry, to the point that I had a hard time reading the text on them.

6. PLOS authors have the option to publish the peer review history of their article (what does this mean?). If published, this will include your full peer review and any attached files.

Reviewer #1: No

Reviewer #2: No

---

## [Author Response · Author response to Decision Letter 0]

13 Dec 2019

We would like to thank the reviewers for their valuable comments. We have answered them in a separate document and added it as 'Response to the Reviewers'.

---

## [Decision Letter · Decision Letter 1]

16 Jan 2020

Extension of a multiphase tumour growth model to study nanoparticle delivery to solid tumours

PONE-D-19-26982R1

Dear Dr. Kremheller,

We are pleased to inform you that your manuscript has been judged scientifically suitable for publication and will be formally accepted for publication once it complies with all outstanding technical requirements.

With kind regards,

Krishna Garikipati, PhD

Academic Editor

PLOS ONE

Additional Editor Comments (optional):

I am pleased to recommend acceptance of your manuscript. Thank you for responding to the reviewers' comments.

Reviewers' comments:

Reviewer's Responses to Questions

**Comments to the Author**

1. If the authors have adequately addressed your comments raised in a previous round of review and you feel that this manuscript is now acceptable for publication, you may indicate that here to bypass the “Comments to the Author” section, enter your conflict of interest statement in the “Confidential to Editor” section, and submit your "Accept" recommendation.

Reviewer #1: All comments have been addressed

Reviewer #2: All comments have been addressed

2. Is the manuscript technically sound, and do the data support the conclusions?

Reviewer #1: Yes

Reviewer #2: Yes

3. Has the statistical analysis been performed appropriately and rigorously? 

Reviewer #1: N/A

Reviewer #2: N/A

4. Have the authors made all data underlying the findings in their manuscript fully available?

Reviewer #1: Yes

Reviewer #2: Yes

5. Is the manuscript presented in an intelligible fashion and written in standard English?

Reviewer #1: Yes

Reviewer #2: Yes

6. Review Comments to the Author

Reviewer #1: The authors have made appropriate revisions that enhance the manuscript. I have no remaining concerns.

Reviewer #2: please, refer to my initial review of the paper and to the response of my comments that the authors provided

7. PLOS authors have the option to publish the peer review history of their article (what does this mean?). If published, this will include your full peer review and any attached files.

Reviewer #1: No

Reviewer #2: No

---

## [Editor Report · Acceptance letter]

23 Jan 2020

PONE-D-19-26982R1

Extension of a multiphase tumour growth model to study nanoparticle delivery to solid tumours

Dear Dr. Kremheller:

I am pleased to inform you that your manuscript has been deemed suitable for publication in PLOS ONE. Congratulations! Your manuscript is now with our production department.

With kind regards,

on behalf of

Prof. Krishna Garikipati

Academic Editor

PLOS ONE